# Biosynthesis of P(3HB-*co*-3HHx) Copolymers by a Newly Engineered Strain of *Cupriavidus necator* PHB^−^4/pBBR_CnPro-*phaC_Rp_* for Skin Tissue Engineering Application

**DOI:** 10.3390/polym14194074

**Published:** 2022-09-28

**Authors:** Chanaporn Trakunjae, Kumar Sudesh, Soon Zher Neoh, Antika Boondaeng, Waraporn Apiwatanapiwat, Phornphimon Janchai, Pilanee Vaithanomsat

**Affiliations:** 1Kasetsart Agricultural and Agro-Industrial Product Improvement Institute (KAPI), Kasetsart University, Chatuchak, Bangkok 10900, Thailand; 2Ecobiomaterial Research Laboratory, School of Biological Sciences, Universiti Sains Malaysia USM, Gelugor 11800, Penang, Malaysia

**Keywords:** polyhydroxyalkanoates, PHA synthase, PhaC, 3-hydroxyhexanoate, *Rhodococcus*, *Cupriavidus necator*, crude palm kernel oil, skin tissue engineering

## Abstract

Polyhydroxyalkanoates (PHAs) are biodegradable polymers synthesized by certain bacteria and archaea with functions comparable to conventional plastics. Previously, our research group reported a newly PHA-producing bacterial strain, *Rhodococcus pyridinivorans* BSRT1-1, from the soil in Thailand. However, this strain’s PHA synthase (*phaC**_Rp_*) gene has not yet been characterized. Thus, this study aims to synthesize PHA using a newly engineered bacterial strain, *Cupriavidus necator* PHB^−^4/pBBR_CnPro-*phaC_Rp_*, which harbors the *phaC**_Rp_* from strain BSRT1-1, and characterize the properties of PHA for skin tissue engineering application. To the best of our knowledge, this is the first study on the characterization of the PhaC from *R. pyridinivorans* species. The results demonstrated that the expression of the *phaC**_Rp_* in *C. necator* PHB^−^4 had developed in PHA production up to 3.1 ± 0.3 g/L when using 10 g/L of crude palm kernel oil (CPKO) as a sole carbon source. Interestingly, the engineered strain produced a 3-hydroxybutyrate (3HB) with 2 mol% of 3-hydroxyhexanoate (3HHx) monomer without adding precursor substrates. In addition, the 70 L stirrer bioreactor improved P(3HB-*co*-2 mol% 3HHx) yield 1.4-fold over the flask scale without altering monomer composition. Furthermore, the characterization of copolymer properties showed that this copolymer is promising for skin tissue engineering applications.

## 1. Introduction

Skin tissue engineering is a promising alternative to replacing conventional dressings and autologous skin transplants for wound-healing purposes [1]. It aims to support skin regeneration when a clinical disorder influences the healing process, for instance, burn injuries, diabetic ulcers, vascular disease, immunocompromised conditions, severe damage, and aging [2]. Presently, a number of bioengineered skin structures have been generated, some of which are commercially available. However, the material used to fabricate scaffolds for engineered skin must possess crucial physical and mechanical properties to keep the wound from fluid loss and contamination [3]. Moreover, it should be easy to handle, biocompatible, biodegradable, and have identical structural and mechanical properties to natural skin.

Polyhydroxyalkanoates (PHAs) are biopolyester synthesized naturally by some bacteria and archaea as an intracellular carbon and energy storage compound. PHAs are attracted particular attention to substitute conventional petrochemical-based plastics owing to their thermal and mechanical properties, which are similar to polypropylene (PP) and polyethylene (PE) [4,5]. In addition, PHAs are a promising biodegradable polymer for making skin tissue scaffolds due to their in vitro biodegradation and tissue compatibility [6,7,8,9,10,11,12]. Among of PHAs members, poly(3-hydroxybutyrate-*co*-3-hydroxyhexanoate) [P(3HB-*co*-3HHx)] copolymer is one of the most attractive copolymers applied in biomedical applications, especially bone and skin tissue engineering applications [3,13]. P(3HB-*co*-3HHx) copolymer, comprising short-chain-length 3-hydroxybutyrate (3HB) and medium-chain-length 3-hydroxyhexanoate (3HHx) monomers, have elastomeric characteristics and decreased crystallinity compared to homogeneous poly(3-hydroxybutyrate), P(3HB) [14].

To date, various bacterial strains have been reported with the ability to synthesize P(3HB-*co*-3HHx) copolymer, including the bacterial strain in the genus *Rhodococcus* [15,16,17,18,19,20]. However, with changing the bacterium and its PHA synthase (PhaC), which is the crucial gene that initiates PHA accumulation in prokaryotes, combined with the carbon sources, polymers with different monomeric compositions and suitable mechanical properties are achieved [21,22]. For instance, Budde et al. (2011) reported two *Ralstonia eutropha* strains that accumulate high levels of P(3HB-*co*-3HHx) from palm oil. These strains express a newly characterized PhaC gene from the strain *Rhodococcus aetherivorans* I24 [18]. Furthermore, Jeon et al. (2014) reported the production of P(3HB-*co*-3HHx) containing up to 40 wt% 3HHx monomers using engineered *R. eutropha* strains comprising deletions of the acetoacetyl-CoA reductase (*phaB*) genes and replacing the native PHA synthase with *phaC2* from *R. aetherivorans* I24 using butyrate as a carbon source [19].

Our group recently isolated a newly bacterial strain of *R. pyridinivorans* BSRT1-1 from the soil in Thailand [23,24]. This strain can produce P(3HB) using fructose as a carbon substrate. However, the PhaC gene of this strain has not been characterized so far. Therefore, this study aims to biosynthesize the PHA using a newly engineered strain of *Cupriavidus necator* PHB^−^4/pBBR_CnPro-*phaC_Rp_*, which replaces the native PhaC with PhaC from *R. pyridinivorans* BSRT1-1, and characterize the properties of the synthesized polymer to apply as a scaffold for skin tissue engineering application. To the best of our knowledge, this is the first study on the characterization of the PhaC gene from *R. pyridinivorans* species.

## 2. Materials and Methods

### 2.1. Bacterial Strains 

All studied bacterial strains and plasmids are listed in Table 1. A bacterial strain of *R. pyridinivorans* BSRT1-1 was grown on Tryptic Soy Agar (TSA) (HiMedia, Maharashtra, India). *C. necator* was cultured on nutrient-rich agar (NR) (10 g/L meat extract, 10 g/L peptone, and 2 g/L yeast extract) with pH 6.8 at 30 °C. *Escherichia coli* was cultured in Luria–Bertani (LB) medium at 37 °C. Kanamycin at a final concentration of 50 µg/ mL was added to the media if necessary. All bacterial strains were stored at 4 °C for further study.

### 2.2. Cloning of PHA Synthase Gene from R. pyridinivorans BSRT1-1

*C. necator* PHB^−^4/pBBR_CnPro-*phaC_Rp_*, an engineered strain of *C. necator* PHB^−^4, was constructed through transconjugating pBBR1MCS-2 plasmid with *phaC1* promotor from *C. necator* (pBBR-CnPro) insert with the PHA synthase gene obtained from *R. pyridinivorans* BSRT1-1(*phaC_Rp_*). The gDNA of *R. pyridinivorans* BSRT1-1 was extracted using the phenol-chloroform technique [25] following the manufacturer’s instructions. The obtained gDNA was used as a template for PCR amplification of *phaC_Rp_*. A forward (Fwd_*phaC_Rp_*) and reverse primer (Rev_*phaC_Rp_*) were designed to amplify *phaC_Rp_*. The pBBR1MCS2 plasmid was used as the shuttle and expression vector, while *C. necator* PHB^–^4, the PHA-negative mutant strain, was used as a host strain. The PCR product was digested with *Hin*dIII and *Apa*I restriction enzymes (Thermo Fisher Scientific, Waltham, MA, USA), and then, it was ligated with *Hin*dIII and *Apa*I-digested pBBR1MCS2 using DNA Ligation Kit Ver.2.1 (Takara Bio Inc., Kusatsu, Japan) according to the manufacturer’s instructions. The resultant recombinant plasmids were transformed into *E. coli* 10G (Lucigen corporation, Middleton, WI, USA), *E. coli* S17-1 and further transconjugated to *C. necator* PHB^−^4 as described by Friedrich et al. (1981) [26]. The successful transformants were verified by colony PCR and DNA sequencing. The DNA sequencing was conducted by Integrated DNA Technologies (IDT) Inc. (Singapore, Republic of Singapore). Finally, the culture of *C. necator* PHB^−^4/pBBR_CnPro-*phaC_Rp_* was stored as glycerol stock in nutrient-rich (NR) broth medium with 30% *v*/*v* of glycerol supplemented with kanamycin (50 µg/mL) at −80 °C. 

**Table 1 polymers-14-04074-t001:** Bacterial strain and plasmids used in this study.

Strain/Plasmid	Description	Reference
*R. pyridinivorans* BSRT1-1	The PHA-producing bacterium isolated from wastewater area soil (Wild-type strain)	[23]
*C. necator* PHB^—^4	The PHA-negative mutant of wild-type *C. necator* H16 (Host strain)	[27]
*E. coli* 10G	The F− *mcr*A ∆(*mrr-hsd*RMS-*mcr*BC) *end*A1 *rec*A1 ϕ80d*lac*Z∆M15 ∆*lac*X74 *ara*D139 ∆(*ara,leu*)7697 *gal*U*gal*K*rps*L (St^rR^) *nup*G λ-*ton*A (Cloning strain)	Lucigen
*E. coli* S17	The *rec*A, *tra* genes of plasmid RP4 integrated into the chromosome, auxotrophic for proline and thiamine (Strain for conjugative transfer of plasmids to host strain)	[28]
pBBR1MCS-2	The Broad-host-range cloning vector Km^r^, *mob*, *lac*Zα (Plasmid)	[29]
pBBR_CnPro	pBBR1MCS-2 plasmid harboring *C. necator phaC1* promotor	[30]
pBBR_CnPro-*phaC_Rp_*	pBBR_CnPro plasmid harboring *phaC2* from *R. pyridinivorans* BSRT1-1 at *Hin*dIII and *Apa*I sites (Plasmid)	This study

### 2.3. PHA Biosynthesis

The biosynthesis of PHA by *C. necator* PHB^−^4/pBBR_CnPro-*phaC_Rp_* was performed using 10 g/L of various carbon sources (Table 2), i.e., fructose, glucose, sucrose, molasses, palm oil (PO), crude palm kernel oil (CPKO), and glycerol. In addition, various precursors or structurally related carbon sources, including, e.g., sodium 4-hydroxybutyrate (Na4HB), Ɣ-butyrolactone (GBL) (Sigma-Aldrich, St. Louis, MO, USA), 1,4 butanediol (BDO) (Merck, Darmstadt, Germany), sodium valerate, and sodium hexanoate (Table 3), were added to the culture medium at a final concentration of 2 g/L to generate the different PHA monomers.

The mineral medium (MM) for PHA biosynthesis was comprised of 4.0 g/L of NaH_2_PO_4_, 4.6 g/L of Na_2_HPO_4_, 0.45 g/L of K_2_SO_4_, 0.54 g/L of urea [CO(NH_2_)_2_], 0.062 g/L of CaCl_2_, 0.39 g/L of MgSO_4_, and 1 mL/L of trace element (TE) solution [18]. The TE solution consisted of ZnSO_4_·7H_2_O, 2.4 g/L; FeSO_4_·7H_2_O, 15 g/L; MnSO_4_·H_2_O, 2.4 g/L; and CuSO_4_·5H_2_O, 0.48 g/L in 0.1 M HCl. The pH of MM was adjusted to 6.8 before sterilization. The carbon sources, precursors, urea, CaCl_2_, MgSO_4_, and TE solution were sterilized separately and added to the MM before bacterial culture inoculation.

The inoculum of *C. necator* PHB^−^4/pBBR_CnPro-*phaC_Rp_* was prepared by transferring three full loops of a single colony in 50 mL of NR medium containing kanamycin (50 µg/mL) and incubated at 30 °C with shaking at 200 rpm for 8 h. After that, the 3% *v*/*v* of prepared inoculum was transferred into the PHA production medium (MM) and incubated at 30 °C with 200 rpm for 48 h. The bacterial cells were collected by transferring the bacterial culture to a 50 mL centrifuge tube and centrifuged at 8000 rpm, 4 °C for 10 min. After that, the supernatant was discarded, and the cell pellets were washed with distilled water. Next, the cell pellets were transferred to a pre-weighed bijoux bottle, then frozen at −20 °C for overnight, and then lyophilized using a freeze dryer. Lastly, the lyophilized cells were weighed to calculate the DCW (g/L). The PHA content and monomer composition of PHA were analyzed by gas chromatography (GC) analysis [31].

### 2.4. Scale-Up in the 70 L Stirred-Tank Bioreactor

The fermentations were carried out in a 70 L stirred-tank bioreactor (B.E. MARUBISHI Co., Ltd., Pathumthani, Thailand) with a working volume of 50 L to improve biomass and P(3HB-*co*-3HHx) yield by *C. necator* PHB^−^4/pBBR_CnPro-*phaC_Rp_*. The seed culture was prepared in NR media supplemented with kanamycin (50 µg/mL). The bioreactor containing MM media was sterilized at 121 °C for 30 min, cooled, and then inoculated with 3% (*v*/*v*) of seed culture. Batch cultivation was performed at 30 °C with a controlled pH of 6.8. A pH controller maintained the culture media by adding 3M of NaOH or H_3_PO_4_. The aeration rate and agitation speed were fixed at 0.25 vvm and 200 rpm, respectively. The PHA content and cell biomass were analyzed at 12 h time intervals for 72 h of fermentation. The fermentation experiments were performed in triplicates, and average values were calculated.

### 2.5. PHA Extraction and Purification

The extraction of PHA accumulated in the cells was performed using the chloroform extraction method [23]. First, 1 g of the freeze-dried cells was dissolved into a stirred chloroform (100 mL) at room temperature. After 3 days of incubation, the cell solution was filtrated through Whatman No. 1 filter paper to eliminate the cell residues. The filtrated solution was then concentrated to approximately 10 mL using a rotary evaporator at a temperature of 50 °C, followed by dropwise adding into a robustly stirred, ice-cold methanol (100 mL). The purified PHA was then harvested and air-dried for 3 days.

### 2.6. PHA Characterization

#### 2.6.1. Proton Nuclear Magnetic Resonance (^1^H-NMR) of Purified PHA

The ^1^H-NMR was performed to confirm PHA’s chemical structure or monomer composition using a Bruker Advance 500 spectrometer (Bruker, Road Billerica, MA, USA) at 30 °C. About 3 mg of the purified PHA was dissolved in 1 mL of deuterated chloroform (CDCl_3_) at a concentration of 25 mg/mL. Tetramethysilane was used as an internal chemical shift reference and NMR analysis was proceeded at 500 MHz with 64 scans. The spectra were calculated for each monomer by comparing with a previous publication.

#### 2.6.2. Thermal Properties of Purified PHA 

The thermal characteristic of the produced polymer was determined using a DSC-60 (Shimadzu, Kyoto, Japan) instrument. The flow rate of nitrogen was fixed at 30 mL/min. Approximately 5 mg of purified PHA was filled into an aluminum pan and heated from 25 to 200 °C with a 15 °C/min heating rate. The heated samples were then maintained at 200 °C for 2 min and subjected to rapid quenching to −40 °C. They were then heated again from −40 to 200 °C at a 15 °C/min heating rate. The glass transition temperature (*T*_g_) and melting temperature (*T*_m_) were observed from the DSC thermogram.

#### 2.6.3. Molecular Weight of Purified PHA

The polymer’s molecular weight was evaluated by gel permeation chromatography (GPC) (10A GPC system, Shimadzu, Kyoto, Japan). About 1 mg purified polymer was dissolved in 1 mL chloroform and analyzed at a column temperature of 40 °C. Polystyrene standards were used to generate a calibration curve. Molecular weight was reported as polystyrene equivalents.

### 2.7. Analytical Methods

The dry cell weight (DCW) of the bacterial cell was determined following the method described by Trakunjae et al. (2021) [23]. The analysis was performed by transferring 1 mL of the cell culture suspension to a pre-weighed Eppendorf tube, added in triplicate, then harvested by centrifugation at 8000 rpm for 10 min. After that, the biomass was washed with distilled water twice. Then, the cell pellet was frozen at −20 °C overnight, followed by lyophilization using a freeze dryer for 48 h. Finally, freeze-dried Eppendorf tubes were weighed again to verify the stability of DCW.

The PHA content and monomers composition of the harvested cell were analyzed by gas chromatography (GC) according to the method described by Braunegg et al. (1978) [31]. The 15–20 mg of freeze-dried cells were weighed in a screw-capped test tube, followed by adding 2 mL of chloroform and 2 mL of methanolysis solution (15% *v*/*v* sulfuric acid and 85% *v*/*v* methanol). The tubes were heated at 100 °C for 140 min and then cooled at room temperature. After that, 1 mL of distilled water was added, then vortexed for 1 min to achieve the layers of phase separation. The lower layer, containing hydroxyacyl methyl esters, was transferred into the tube containing sodium sulfate anhydrous to eliminate residue water. Then, the 500 mL of hydroxyacyl methyl esters solution and 500 mL of 0.2% *v*/*v* of caprylic acid methyl ester (CME) solution were mixed in a GC vial. In this case, the CME solution was used as an internal standard. The sample was analyzed by GC2014 (Shimadzu, Kyoto, Japan) equipped with an AOC-20i auto-injector (Shimadzu, Kyoto, Japan), Restek RTX-1 capillary column, and a flame ionization detector (FID). The injector temperature was set at 270 °C, and the temperature of column was set at 70 °C and slowly increased at a rate of 10 °C/min to 280 °C, while the detector temperature was established at 280 °C. Nitrogen was used as a carrier gas.

### 2.8. Statistical Analysis

All experiments were performed in triplicates. The obtained experimental data are indicated as the mean value of the triplicate with standard error.

## 3. Results and Discussion

### 3.1. Characterization of the PHA Synthase Gene 

PhaC is the crucial enzyme for the synthesis of PHA, where the monomers are polymerized of PHA into PHA polymers. The substrate specificity of the PhaC determines the types of monomers incorporated into the PHA polymer [32,33]. In this study, the nucleotide and amino acid sequence of *phaC_Rp_*, a newly isolated PHA-producing strain, was obtained and used for the initial BLAST search. The nucleotide and amino acid length of *phaC_Rp_* were 1755 bp and 584 amino acids, respectively. According to the BLASTn analysis, the *phaC_Rp_* revealed significant similarity with the *phaC* of *R. rodochrous* ATCC BAA870 (CP032675.1) (97.60%), *R. rodochrous* EP4 (CP032221.1) (97.60%), *R. pyridinivorans* DNHP-S2 (CP088975.1) (95.20%), *R. pyridinivorans* YC-MTN (CP095851.1) (95.20%), and *R. pyridinivorans* SB3094 (CP006996.1) (94.85%), respectively. The determination of the taxonomic position of *phaC_Rp_* was carried out by the construction of phylogenetic analysis to compare its amino acid sequence with that of other bacterial species. The *phaC_Rp_* formed a coherent clade with strain *R. pyridinivorans* KG-16 (CAA47035.1) in the NJ phylogenetic tree (Figure 1). In addition, the *phaC_Rp_* formed a cluster with the *phaC* of *R. pyridinivorans* SB3094 (AHD20144.1), *R. aetherivorans* I24 (*phaC1*) (AED02507), and *R. ruber* (CAA47035.1), respectively.

PhaCs have been classified into four classes according to their primary sequences, substrate specificity, and subunit composition [33]. The class I, III, and IV PhaCs desire short-chain length monomers containing C3-C5 carbon chain lengths, such as 3-hydroxybutyrate (3HB). In contrast, class II PhaCs exhibit higher activities toward the monomers with medium-chain length comprising C6-C14 carbon chain lengths, such as the C6 monomer 3-hydroxyhexanoate (3HHx). However, PHA produced from short-chain-length and medium-chain-length monomer combinations has superior thermal and physical properties compared to the homopolymers. [33,34,35]. In 1999, Rehm and Steinbuchel reported that class I and class II PhaCs include enzymes containing only one type of subunit with molecular weights (*M*_W_) between 61 and 68 kDa [36]. Furthermore, six conserved blocks of amino acid sequences were identified, revealing areas of higher similarity. In contrast, the N-terminal region, approximately 100 amino acids relative to type I PhaCs, is highly variable [37]. In this study, the NJ phylogenetic tree showed that the *phaC_Rp_* is a consistent clade with the PhaC of those bacteria in class I PHA synthases (Figure 1). Similarly, the *phaC* from *R. ruber* is encoding for a class I PHA synthase [37]. In addition, the gene coding for a putative PHA depolymerase (PhaZ), which is expected to be the enzyme involved in PHA mobilization, was reported downstream of the *R. ruber phaC*. In addition, *R. aetherivorans* revealed two putative class I PhaC genes identified as *phaC1_Ra_* and *phaC2_Ra_* [18]. Among the *Rhodococcus* species, *R. aetherivorans* I24 was extensively studied for PHA production [18]. When comparing the amino acid sequences of *phaC_Rp_* with *phaC1_Ra_* and *phaC2_Ra_,* it was shown that the *phaC_Rp_* and *phaC1_Ra_* have very analogous sequences arrangement, which is 80% similar, while showing 39% similarity with the *phaC2_Ra_*.

### 3.2. PHA Biosynthesis and Heterologous Expression of phaC_Rp_ in C. necator PHB^–^4

PHA biosynthesis was performed in a 250 mL flask to evaluate PHA production by the newly engineered strain of *C. necator* PHB^−^4/pBBR_CnPro-*phaC_Rp_* and to select the best carbon and precursors for further studies. *C. necator* PHB^−^4/pBBR_CnPro-*phaC_Rp_* was cultured under nitrogen-limiting conditions at 30 °C for 48 h using different carbon sources and precursor or structurally related carbon sources. Of seven carbon sources, i.e., glucose, fructose, sucrose, glycerol, molasses, crude palm kernel oil (CPKO), and palm oil, CPKO was found to be the best carbon source for cell growth and PHA biosynthesis by *C. necator* PHB^−^4/pBBR_CnPro-*phaC_Rp_* (Table 2). Approximately 6.3 ± 0.4 g/L of DCW and 49 wt%DCW of PHA content were achieved when using CPKO as a carbon source, whereas 3.7 ± 0.3, 2.1 ± 0.2, 0.5 ± 0.0, 0.4 ± 0.0, and 0.3 ± 0.0 g/L of DCW were obtained when using PO, fructose, molasses, glycerol, glucose, and sucrose as a carbon source, respectively. In addition, the PHA contents of 41 ± 3, 36 ± 2, 30 ± 5, 1.1 ± 0, and 0.6 ± 0 wt%DCW were achieved when using PO, fructose, molasses, glycerol, and sucrose as a carbon substrate. However, this strain was not able to accumulate the PHA content when using glucose as a carbon source. Interestingly, the 2 mol% of 3HHx monomers were successfully incorporated by *C. necator* PHB^−^4/pBBR_CnPro-*phaC_Rp_* without adding any precursor or structurally related carbon sources. Nevertheless, the low incorporation of 3HHx monomers fraction might be due to the high intracellular concentrations of 3HB-CoA, which limit 3HHx incorporation into the PHA produced by the *C. necator* PHB^−^4/pBBR_CnPro-*phaC_Rp_* [18].

Plant oils have been reported as an excellent carbon source for PHA production by many bacterial strains. For example, the biosynthesis of P(3HB-*co*-3HHx) directly from palm oil has been reported by Budde et al. (2011) using recombinant *Ralstonia eutropha* strains expressing heterologous PhaCs from the bacterium *R. aetherivorans* I24 [18]. *R. aetherivorans* I24 was extensively studied for PHA production among the *Rhodococcus* species. Budde et al. [18] reported the production of P(3HB-*co*-3HHx) by *R. eutropha* (*C. necator*) Re2000 and Re2001, which were engineered bacterial strains containing *phaC1_Ra_* and *phaC2_Ra_*, respectively. The results revealed that strains Re2000 and Re2001 produced 5.98 and 1.09 g/L of P(3HB-*co*-3HHx) incorporated with 1.1 ± 0.3 and 1.5 ± 0.2 mol% when using palm oil as a sole carbon source, respectively. The P(3HB-*co*-3HHx) production from strain Re2000 was slightly higher than in this study, in which *C. necator* PHB^-^4/pBBR_CnPro-*phaC_Rp_* obtained 3.1 g/L of P(3HB-*co*-3HHx). However, this strain produced a molar fraction of 2 mol% 3HHX incorporated in P(3HB-*co*-3HHx), which was higher than those produced by strains Re2000 and Re2001.

Furthermore, the production of P(3HB-*co*-3HHx) copolymer including high 3HHx monomer fraction by *C. necator* strain Re2160/pCB113 was studied using CPKO as a carbon substrate [38]. In addition, the combinations of CPKO and oil palm tree trunk sap (OPTS) as a carbon substrate for the biosynthesis of P(3HB-*co*-3HHx) using the recombinant *C. necator* strain Re2058/pCB113 were reported by Muragan et al. (2016) [39]. Therefore, this study selected CPKO as the carbon source for further experiments.

Apart from carbon sources, several types of precursors or structurally carbon substrates were tested to enhance the P(3HB-*co*-3HHx) production and improve the ability of *C. necator* PHB^−^4/pBBR_CnPro-*phaC_Rp_* for synthesizing different types of PHA monomers. The 2 g/L of five different precursors, i.e., sodium 4-hydroxybutyrate, Ɣ–Butyrolactone, sodium valerate, 1,4-Butanediol, sodium heptanoate, and sodium hexanoate, were supplemented in the MM. The fermentations were carried out in a 250 mL flask at 30 °C for 48 h, using 10 g/L of CPKO as the primary carbon substrate. The results revealed that 3HHx monomers were successfully incorporated by *C. necator* PHB^−^4/pBBR_CnPro-*phaC_Rp_* in all experimental runs (Table 3). Furthermore, as high as 4 mol% of 3HHx monomer was successfully produced by *C. necator* PHB^−^4/pBBR_CnPro-*phaC_Rp_* when co-fed with 2 g/L of sodium hexanoate. Based on this result, the 3HHx fraction was calculated as two times higher than it would have been without adding sodium hexanoate. It has been proven that if a PhaC able to polymerize MCL monomers is expressed in recombinant *R. eutropha*, the strain will accumulate MCL PHA when cultivated in fatty acids [18,40]. However, this strain was not able to synthesize other types of PHA monomer such as 3-hydroxyvalerate (3HV) and 4-hydroxybutyrate (4HB), even when co-feeding with sodium valerate, sodium 4-hydroxybutyrate, Ɣ–Butyrolactone, and 1,4-Butanediol. Moreover, the result shows that the addition of precursors decreased cell biomass.

To compare the PHA production ability of an engineered strain with the wild-type strain, the PHA biosynthesis of *C. necator* H16, *C. necator* PHB¯4, *R. pyridinivorans* BSRT1-1, and *C. necator* PHB^−^4/pBBR_CnPro-*phaC_Rp_* were performed using 10 g/L CPKO as sole carbon source without the addition of precursors. The results showed that the *C. necator* PHB^−^4/pBBR_CnPro-*phaC_Rp_* produced 3.1 g/L of total PHA containing 2 mol% of 3HHx fraction. Meanwhile *R. pyridinivorans BSRT1-1*, which is the wild-type strain, and *C. necator* PHB^—^4, which is the negative control strain, cannot produce PHA when using CPKO as the sole carbon source (Table 4). However, compared to the well-studied PHA-producing strain *C. necator* H16, *C. necator* PHB^−^4/pBBR_CnPro-*phaC_Rp_* produced less PHA. Nevertheless, this result revealed that with modifying the bacterium and its PhaC, combined with the carbon sources, polymers with different monomeric compositions are obtained [21,22].

### 3.3. Scaling Up P(3HB-co-3HHx) Production in a 70 L Stirred-Tank Bioreactor

The enhancement of P(3HB-*co*-3HHx) production by *C. necator* PHB^−^4/pBBR_CnPro-*phaC_Rp_* was carried out with a batch cultivation in a 70 L stirred-tank bioreactor containing 50 L of MM. A total of 10 g/L of CPKO was applied as a sole carbon source. The temperature, pH, aeration rate, and agitation speed were fixed at 30 °C, 6.8 and 0.25 vvm, and 200 rpm, respectively. During the first 48 h of fermentation, the growth of *C. necator* PHB^−^4/pBBR_CnPro-*phaC_Rp_* gradually increased, then remained constant until 60 h; after that, it slightly decreased at 72 h (Figure 2). Similarly, the PHA accumulation was increased in the bioreactor during the 48 h of fermentation but significantly decreased at 72 h (Figure 2).

The highest production of P(3HB-*co*-3HHx) was at 48 h when the biomass, PHA content, and 3HHx mol% were 7.7 ± 0.6 g/L, 56 ± 5 wt%DCW, and 2 mol%, respectively (Figure 2). According to the results, the biomass of *C. necator* PHB^−^4/pBBR_CnPro-*phaC_Rp_* in a 70 L stirred-tank bioreactor improved by 1.4-fold compared with the shake flask scale. However, this approach cannot increase PHA accumulation and the mol% fraction of 3HHx. This result might be due to the limitation of single-batch production [21,41]. Batch cultivations for manufacturing PHA are straightforward to operate but intrinsically limited productivity. The highest allowed concentration of nitrogen and carbon supply at the beginning of the fermentation batch is limited by the physiological criteria of the production strain. Nevertheless, the development of PHA production using batch fermentation by various types of bacteria has been reported [42].

Previously, Tanaka et al. [43] reported the production of P(3HB-*co*-3HHx) from CO_2_ by engineered *C. necator* MF01∆B1/pBBP-ccrMeJ4a-emd. These strains can synthesize P(3HB-*co*-3HHx) containing 47.7 ± 6.2 mol% of 3HHx from fructose using a complete mineral medium and a substrate gas mixture (H_2_/O_2_/CO_2_). However, the composition of 3HHx in P(3HB-*co*-3HHx) was controlled to 11 mol%, which is more suitable for practical applications. Furthermore, Thinagaran and Sudesh [44] have studied the biosynthesis of P(3HB-*co*-21 mol% 3HHx) using emulsified sludge palm oil by an engineered *C. necator* Re2058/pCB113. The 9.7 g/L of dried cells mass of *C. necator* Re2058/pCB113 with 74 wt% of PHA content was obtained [45]. Furthermore, Riedel et al. successfully synthesized 49–72 wt% of P(3HB-*co*-3HHx), which incorporated 16–27 mol% of 3HHx composition using different waste animal fats by *C. necator* Re2058/pCB113 [45].

### 3.4. Characterization of P(3HB-co-3HHx) produced by C. necator PHB^−^4/pBBR_CnPro-phaC_Rp_

The ^1^H NMR was performed to confirm 3HB and 3HHx units in the polymer fraction of PHA copolymer synthesized by the strain *C. necator* PHB^−^4/pBBR_CnPro-*phaC_Rp_*. Figure 3 shows the ^1^H NMR spectrum of H4 and H6, which represent the C4 methylene groups and C6 methyl-group in the 3HHx unit, respectively, implying the actual formation of the P(3HB-*co*-3HHx) copolymer [37,43]. However, the signal of H4 and H6 appears indistinct due to the low mol% fraction of 3HHx.

In this study, the molecular weight of the P(3HB-*co*-2 mol% 3HHx) copolymers produced by *C. necator* PHB^−^4/pBBR_CnPro-*phaC_Rp_* was analyzed by gel permeation chromatography (GPC). The weight-average molecular weight (*M*_w_) of 6.27 × 10^5^ Da was obtained with the polydispersity (*M*_w_/*M*_n_) of 1.70. Typically, the molecular weight of the PHA varies for several reasons, for instance, the availability of precursors for PHA synthesis, the type of PhaC, the expression level of PhaCs synthases, and the availability of enzymes that hydrolyze PHA [34].

The molecular weight of P(3HB-*co*-3HHx), which has a high 3HHx monomer percentage, is typically significantly lower than that of P(3HB) homopolymer synthesized by wild-type *C. necator* H16. [46]

Thermal properties of P(3HB-*co*-3HHx) copolymer synthesized by *C. necator* PHB^−^4/pBBR_CnPro-*phaC_Rp_* were analyzed using DSC. Defining the polymer thermal stability is essential for understanding the chemical recycling of polymer materials [38]. Values for DSC analysis were recorded from the second heating to eradicate the thermal history of the previous samples. The melting temperature (*T*_m_) and glass transition temperature (*T*_g_) of P(3HB-*co*-3HHx) were 160.18 and 6.04 °C, respectively (Figure 4). The melting temperature of P(3HB-*co*-3HHx) copolymers are commonly lesser than those of P(3HB). However, there was no significant correlation between these parameters and 3HHx molar fractions [47].

PHAs have been reported for application for wound repair and skin tissue engineering applications [48,49,50,51]. The chemical properties that require consideration involve lowering the melting point and glass transition temperature. These characteristics rely upon the polymer’s monomeric composition and molecular weight. PHA copolymers with high molecular weights can overcome these weaknesses [48]. However, after implantation, the main concern is the outcome of the products from PHA degradation and whether the PHA residuals are biocompatible with the cells at locations other than where they were proposed [49]. Among members of the PHAs family, P(3HB-*co*-3HHx) are promising biomaterials for the application in tissue engineering because they are biodegradable, biocompatible, and low cytotoxic.

In this study, P(3HB-*co*-2 mol% 3HHx) copolymer was produced by a newly engineered strain of *C. necator* PHB^−^4/pBBR_CnPro-*phaC_Rp_*, which possesses physical and chemical properties to be applied for wound repair or skin tissue engineering application.

Previously, Li et al. (2008) studied the blending of P(3HB)/P(3HB-*co*-3HHx) and P(3HB)/P(3HB-*co*-4HB) copolymers in several ratios for fabricating the nanofiber 3D scaffolds and matrices cast solution for testing with human epidermal cells (keratinocytes) [50]. Investigations of keratinocyte attachment and proliferation have shown that the significant promotion of cell adhesion and proliferation could be obtained by fibrous matrices. There was a striking resemblance between the porosity structure and the collagen ECM organization. This study proved that the PHA-based matrices have the potential for application in skin tissue engineering [50]. Furthermore, Tang et al. (2008) fabricated the nanostructured fibrous scaffolds of poly(3HB-*co*-5 mol% 3HHx), poly(3HB-*co*-7 mol% 4HB), and poly(3HB-*co*-97 mol% 4HB) copolymers [51]. They found that the tensile strength and Young’s modulus of these scaffolds were comparable to those of human skin. The histological analysis confirmed the electrospun PHA scaffolds were well-tolerated in vivo after subcutaneous implantation [51].

## 4. Conclusions

This is the first study on the characterization of PhaC of *R. pyridinivorans* species. The strain *C. necator* PHB^−^4/pBBR_CnPro-*phaC_Rp_*, a newly engineered strain of *C. necator* PHB^−^4 with *phaC_Rp_*. CPKO and sodium hexanoate were found to be the best carbon source and precursor for P(3HB-*co*-3HHx) copolymer production by this strain, respectively. Under fermentation conditions, this strain can accumulate 56 ± 5 wt%DCW of PHA containing 2 mol% of 3HHx and produce 7.7 ± 0.6 g/L of biomass using a 70 L stirred-tank bioreactor. Furthermore, the properties of the P(3HB-*co*-3HHx) copolymer have demonstrated that this copolymer is promising for skin tissue engineering applications.

## Figures and Tables

**Figure 1 polymers-14-04074-f001:**
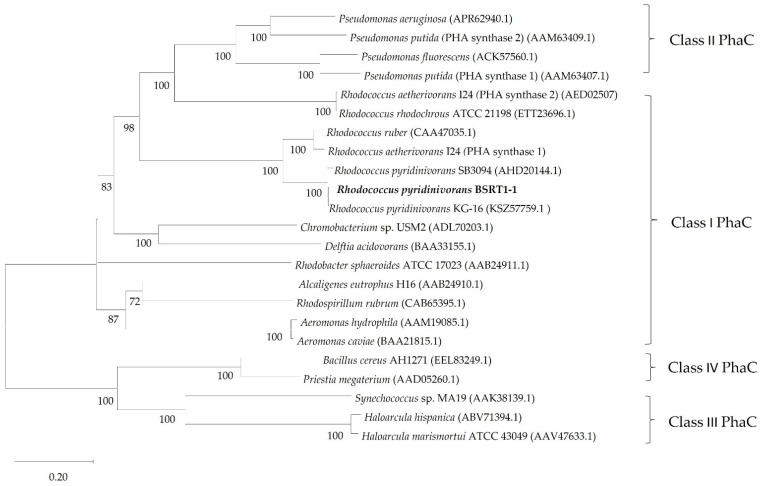
Neighbor-joining phylogenetic tree of PhaC amino acid sequence of *R. pyridinivorans* BSRT1-1 and closely related taxonomic group using amino acid sequence achieved from GenBank database. Numbers at nodes indicate levels of bootstrap support (%) according to the neighbor-joining analysis of 1000 resampled datasets; only values  ≥ 50% are given. Bar, 0.20 substitutions per site.

**Figure 2 polymers-14-04074-f002:**
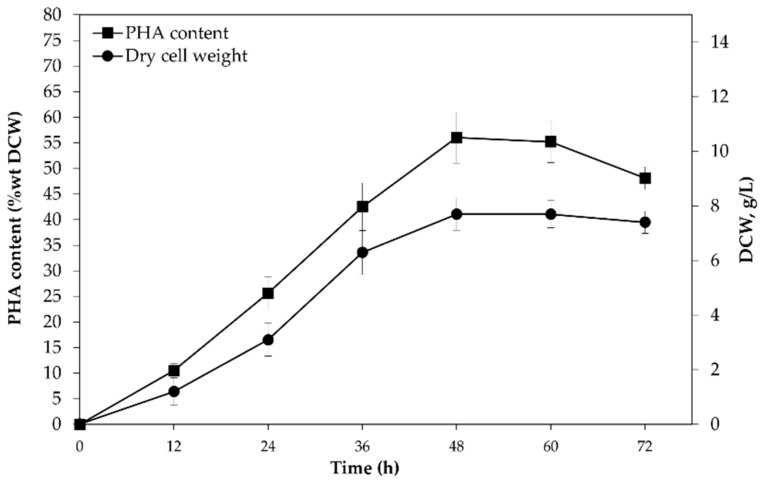
PHA production and biomass of *C. necator* PHB^−^4/pBBR_CnPro-*phaC_Rp_* in 70 L stirred-tank bioreactor during 72 h.

**Figure 3 polymers-14-04074-f003:**
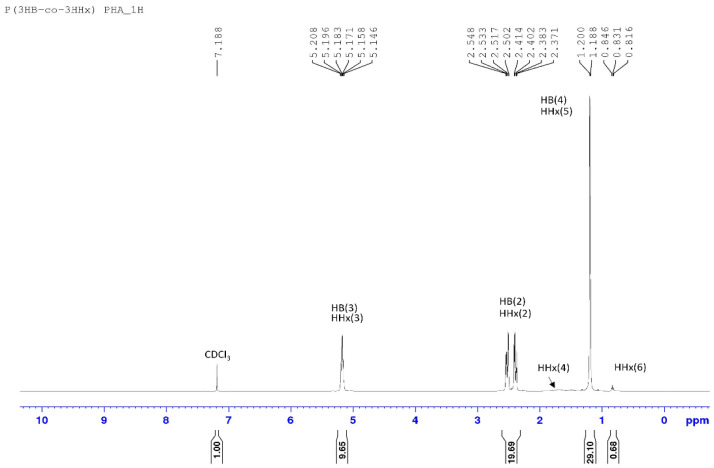
Proton Nuclear Magnetic Resonance Spectroscopy (^1^H NMR) spectrum of P(3HB-*co*-2 mol% 3HHx) produced by *C. necator* PHB^−^4/pBBR_CnPro-*phaC_Rp_*.

**Figure 4 polymers-14-04074-f004:**
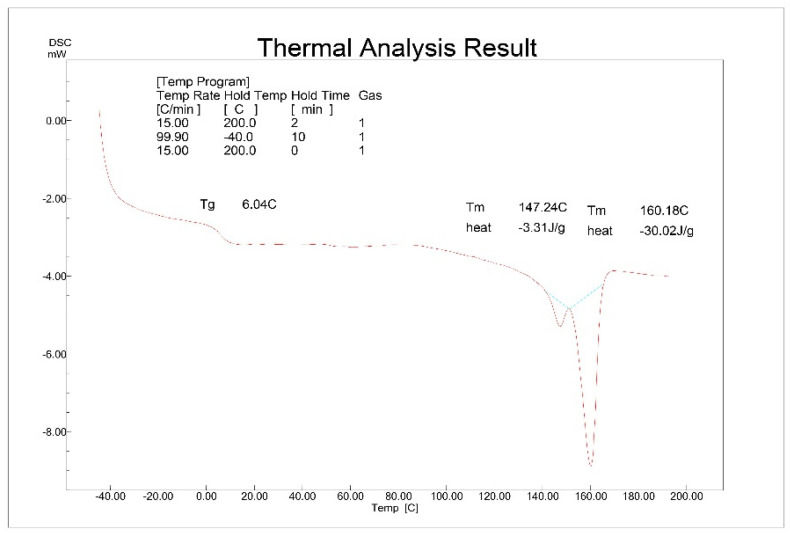
Differential Scanning Calorimetry (DSC) of P(3HB-*co*-2 mol% 3HHx) produced by *C. necator* PHB^−^4/pBBR_CnPro-*phaC_Rp_*.

**Table 2 polymers-14-04074-t002:** PHA biosynthesis by *C. necator* PHB^−^4/pBBR_CnPro-*phaC_Rp_* using different carbon sources in shake flask experiments.

Carbon Sources (10 g/L)	DCW	PHA Content	Monomer Composition (mol%)			
	(g/L)	(wt%DCW)	3HB	3HV	3HHx	4HB
Glucose	0.4 ± 0	0	ND	ND	ND	ND
Fructose	3.7 ± 0.3	36 ± 2	100 ± 0	-	-	-
Sucrose	0.3 ± 0	0.6 ± 0	ND	ND	ND	ND
Sugarcane molasses	2.1 ± 0.2	30 ± 5	100 ± 0	-	-	-
Palm oil (PO)	5.7 ± 0.5	41 ± 3	100 ± 0	-	-	-
Crude palm kernel oil (CPKO)	6.3 ± 0.4	49 ± 1	98 ± 1	-	2 ± 0	-
Glycerol	0.5 ± 0	1.1 ± 0	ND	ND	ND	ND

ND, not determined.

**Table 3 polymers-14-04074-t003:** PHA biosynthesis by *C. necator* PHB^−^4/pBBR_CnPro-*phaC_Rp_* using different precursors in shake flask experiments.

Precursor (2 g/L)	DCW	PHA Content	Monomer Composition (mol%)			
	(g/L)	(wt%DCW)	3HB	3HV	3HHx	4HB
Sodium 4-hydroxybutyrate	5.5 ± 0.2	44 ± 5	98 ± 1	-	2 ± 0	-
Ɣ–Butyrolactone	5.2 ± 0.5	45 ± 2	98 ± 1	-	2 ± 0	-
1,4-Butanediol	5.0 ± 0.3	42 ± 2	99 ± 1	-	1 ± 0	-
Sodium valerate	5.6 ± 0.1	48 ± 3	98 ± 1	-	2 ± 0	-
Sodium hexanoate	5.8 ± 0.3	54 ± 2	96 ± 1	-	4 ± 0	-

**Table 4 polymers-14-04074-t004:** Comparison of PHA biosynthesis by *C. necator* H16, *C. necator* PHB^—^4, *R. pyridinivorans* BSRT1-1, and *C. necator* PHB^−^4/pBBR_CnPro-*phaC_Rp_* using 10 g/L CPKO as sole carbon sources.

Strain	DCW	PHA Content	PHA Concentration	Monomer Composition (mol%)			
	(g/L)	(wt%DCW)	(g/L)	3HB	3HV	3HHx	4HB
*C. necator* H16	6.7 ± 0.3	79 ± 2	5.3	100 ± 0	-	-	-
*C. necator* PHB¯4	0.3 ± 0	0	ND	ND	ND	ND	ND
*R. pyridinivorans* BSRT1-1	0.2 ± 0	0	ND	ND	ND	ND	ND
*C. necator* PHB^−^4/pBBR_CnPro-*phaC_Rp_*	6.3 ± 0.4	49 ± 1	3.1	98 ± 1	-	2 ± 0	-

ND, not determined.

## Data Availability

Data are contained within the article.

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
