# Peer review of "Biosynthesis of P(3HB-*co*-3HHx) Copolymers by a Newly Engineered Strain of *Cupriavidus necator* PHB^−^4/pBBR_CnPro-*phaC_Rp_* for Skin Tissue Engineering Application"

_polymers, 2022, doi:10.3390/polym14194074_

Round 1

Reviewer 1 Report

This work by Trakunjae et al. studied a biodegradable polymer synthesized by an engineered strain of a specific bacterium.  The authors extracted polyhydroxyalkanoates biosynthesized by C. necator PHB-4 with several carbon sources and precursors.  The chemical compositions and thermal properties of the polymer materials were characterized by NMR spectrometry and differential scanning calorimetry.

                The manuscript is well written and can be a good contribution to the field of bioengineering.  I will recommend its publication after the authors address the following points as a major revision.

1.     There are already several works published on Rhodococcus aetherivorans.  However, no comparisons were made in the manuscript, which made it hard to see what is advantageous in using Rhodococcus pyridinivorans.

2.     On page 10, it is stated that the copolymer had properties good enough for wound repair/ skin tissue engineering applications.  This claim needs to be supported by data.  For instance, the authors should measure the mechanical properties of the copolymers.

3.     In Figure 2, I cannot find the “exponential phase” of the bacterial growth. 

Author Response

Point 1: There are already several works published on Rhodococcus aetherivorans.  However, no comparisons were made in the manuscript, which made it hard to see what is advantageous in using Rhodococcus pyridinivorans.

Respose to Point 1: The comparison of the PHA synthase gene of R. aetherivorans (phaCRa) and R. pyridinivorans (phaCRp), used in this study, was added to this manuscript, as highlighted in section 3.1 on page 6 (line no 245-248) and page 7 (line no 282-290). 

Point 2: On page 10, it is stated that the copolymer had properties good enough for wound repair/ skin tissue engineering applications.  This claim needs to be supported by data.  For instance, the authors should measure the mechanical properties of the copolymers.

Response to Point 2: Agreed, since we did not perform the mechanical properties experiment, we edited the sentence as stated in the manuscript to avoid overclaiming our produced polymer for wound repair/skin tissue engineering application. Kindly check the edited version as highlighted in section 3.3 on page 10 (line no 417-419).

Point 3: In Figure 2, I cannot find the “exponential phase” of the bacterial growth.

Response to Point 3: According to your comments, we changed the descriptions of Figure 2, as highlighted in section 3.3 on page 8 (line no 344-346). 

Reviewer 2 Report

PHA is one of the essential biodegradable polymers. This study provides an interesting insight into the biosynthesis of PHA.

1) Title: This study investigates the physical properties of P(3HB-co-3HHx) but not its skin tissue engineering application. It only infers its use by analogy from its physical properties obtained. I have my doubts about including "skin tissue engineering application" in the title of this paper.

2) Table 2 is missing.

3) R. oyridinivorans BSRT1-1 is not able to make PHA using CPKO. Why is the introduction of pBBR1MCS-2 phaCRp into C. necator PHB-4 able to biodegrade palm oil and CPKO? The characteristics are significantly changed. Please explain the reason for this.

4) C. necator H16 is a well-known strain in this field. Please compare PHA production in studies in which genes from these bacteria have been introduced into other bacteria.

Author Response

Point 1: Title: This study investigates the physical properties of P(3HB-co-3HHx) but not its skin tissue engineering application. It only infers its use by analogy from its physical properties obtained. I have my doubts about including "skin tissue engineering application" in the title of this paper.

Response to Point 1: The main focus of this study was to characterize the PHA synthase gene of R. pyridinivorans BSRT1-1 to biosynthesize P(3HB-co-3HHx) copolymer for skin tissue engineering application. Although, we did not study the compatibility of produced P(3HB-co-3HHx) copolymer with the skin cells, the physical and chemical properties of the produced copolymer in this study demonstrated that this copolymer is promising for skin tissue engineering applications. Therefore, the addition of "skin tissue engineering application" in the title of this manuscript is crucial for clarifying the benefit of the polymer in medical applications.

Point 2:  Table 2 is missing.

Response to Point 2: There is a mistake in Table arrangement. There is no Table 2 in this manuscript.  As highlighted in the table caption, we changed the Table order number.

Point 3: R. pyridinivorans BSRT1-1 is not able to make PHA using CPKO. Why is the introduction of pBBR1MCS-2 phaCRp into C. necator PHB-4 able to biodegrade palm oil and CPKO? The characteristics are significantly changed. Please explain the reason for this.

Response to Point 3: In this case, the characteristic change was influenced by host strains, not phaCRp. This study used C. necator PHB4 as a host strain. It is a PHA-negative mutant of wild-type C. necator H16, a well-studied PHA producer. This strain can grow well in various types of oil, including PO and CPKO, through the β-oxidation pathway.

Point 4:  C. necator H16 is a well-known strain in this field. Please compare PHA production in studies in which genes from these bacteria have been introduced into other bacteria.

Response to Point 4: According to your comment, several studies have been conducted on C. necator as a host strain for P(3HB-co-3HHx) copolymers production, including other types of PHA copolymers. However, a few studies of transformed phaC of C. necator H16 to other bacteria have been reported because this phaC only expresses scl-PHA, for example, P(3HB), which has poor property to apply for various applications. However, we have added the comparison of PHA production of recombinant C. necator with other bacteria, as highlighted in section 3.3 on page 9 (line no 363-373).

Round 2

Reviewer 1 Report

The authors gave answers to all of the comments raised.  I now recommend publication of this manuscript.

Reviewer 2 Report

The revisions were well done. This paper is acceptable.